# Effects of a Smoking Cessation Counseling Education Program on Nursing Students

**DOI:** 10.3390/healthcare11202734

**Published:** 2023-10-13

**Authors:** Sung-Rae Shin, Eun-Hye Lee

**Affiliations:** Nursing Department, College of Nursing, Sahmyook University, Seoul 01795, Republic of Korea; shinsr@syu.ac.kr

**Keywords:** smoking cessation, counseling, education, undergraduate (students), nursing

## Abstract

This is a quasi-experimental study applying a nonequivalent control group pre-test–post-test design with the control of exogenous variables to compare the research variables among experimental groups. Participants totaled 67 and were divided into three groups, each participating in a different training program (lecture, online video, and case-based peer role-play). There were significant increases in attitudes toward smoking cessation interventions in Experiment 2 (online video) (t = −2.48, *p* = .021) and Experiment 3 (case-based peer role-play) (t = −2.69. *p* = .013), efficacy of smoking cessation interventions in Experiment 2 (−2.06, *p* = .052), and intention to deliver smoking cessation intervention in all experimental groups (Exp 1 t = −5.54, *p* < .001; Exp 2 t = −2.83, *p* = .010; Exp 3 t = −3.50, *p* = .002). All three programs of smoking cessation counseling education (lecture, online video, and case-based peer role-play) used in this study showed meaningful results on the study variables. In conclusion, all of the approaches of this study were found to be effective on the intention to deliver smoking cessation intervention, and it is important to creatively apply counseling programs that include essential elements of smoking cessation interventions in nursing education settings.

## 1. Introduction

Smoking is associated with lung cancer and numerous chronic diseases of the respiratory, cardiovascular, and digestive systems, and it is one of the most preventable causes of premature death and disability [1]. According to the statistics on the causes of death in Korea in 2020, bronchial and lung cancers accounted for 18,673 cases, ranking first, and its mortality rate is continuously increasing. Additionally, the number of deaths due to acclimatization, chronic respiratory diseases, and diabetes is rapidly increasing [2]. Considering the relationship between these diseases and smoking, smoking cessation is an important factor in improving patients’ health and quality of life.

According to the US Department of Health Clinical Guidelines for the Treatment of Tobacco Use and Addiction (Clinical Practice Guideline, Treating Tobacco Use and Dependence, 2008), when all healthcare providers (e.g., doctors, nurses, dentists, pharmacists, and respiratory technicians) meet patients who smoke, they are required to assess and record whether they smoke, and provide information related to smoking cessation [3]. To increase the effect of smoking cessation intervention, it has been found that an increase in the number of counseling sessions, securing the number of appropriately trained medical personnel on smoking cessation, the length of counseling time, and the diversity of healthcare professionals involved in counseling increases the participants’ smoking cessation rate [3,4,5].

Nurses, as professional healthcare providers, who spend considerable time caring for patients and being close to them in various treatment processes, have the potential to greatly influence and reduce smoking rates through smoking cessation counseling and education. In contrast with the fact that 91% of nursing schools in the UK provide smoking cessation intervention education in their curriculum, nursing colleges in Korea provided less than one hour of smoking cessation education in their curriculum [6]. Notably, according to a study that surveyed clinical nurses across the country, a lack of skills, confidence, and knowledge were identified as major obstacles to patient smoking cessation interventions [7]. Most of the 23,000 students trained annually in nursing colleges in Korea come into contact with patients who smoke in clinical settings; hence, an educational strategy to help them actively engage in smoking cessation interventions is crucial.

The theory of planned action suggests action intention as an important predictor of the performance of a specific action, and it is known that such action intention is influenced by beliefs and attitudes toward actions and self-efficacy factors [8]. In addition, research on intentions to deliver smoking cessation interventions has shown that nurses’ attitudes toward tobacco cessation influence their intentions to provide cessation advice when it is desirable, and that self-efficacy in being able to provide tobacco cessation interventions is a significant influence on nurses’ intentions to provide smoking cessation interventions [9,10]. However, there is a lack of educational content or programs to enhance the competency of these nurses so that they can effectively implement smoking cessation interventions.

Lecture is the most commonly used and known as the cost-effective method of teaching in nursing education [11,12]. Recently various online education program has been developed and are widely used in nursing education because they allowed for repeated learning based on the needs of the students [13,14]. Among the various educational methods used in nursing education, peer role-play is a technique that is vital in situations based on a case. Empathy can be improved by playing the role of a smoker or smoking cessation counselor, or by sharing experiences [15,16]. The problem-solving process of the participant through peer role-play can improve verbal and non-verbal communication skills in the process of interaction between students from the performance to the evaluation process, and is more cost-effective than using standardized patients [15,17,18]. In this a study, lecture, online video, and case-based peer role-play approach was used to apply a smoking cessation counseling education to nursing students to identify changes in nursing students’ attitudes toward smoking cessation interventions, self-efficacy, and intention to deliver smoking cessation interventions. We aimed to contribute to the development of smoking cessation counseling educational content that can be used for undergraduate nursing curricula.

## 2. Research Methods

### 2.1. Research Design

This quasi-experimental study design applied a non-equivalent control group pre-test–post-test design with the control of exogenous variables (Figure 1).

### 2.2. Research Participants

The subject of this study was a convenience sampling targeting third year nursing students enrolling in the nursing department located in S city. The research intervention consisted of a group receiving a pre-determined 6-week clinical practice on Fridays when there was no hospital clinical practice. The first two weeks were assigned to experimental group 1 (lecture group), the next two weeks to experimental group 2 (online video group), and the last two weeks to experimental group 3 (case-based peer role-play group) to prevent the spread of the intervention effect.

Our calculation showed that at least 51 participants were required based on the following: an effect size, .50 [19]; significance level (α), .05; test power (1-β), .95; number of time points, 2; number of group, 3. A repeated measure analysis of variance (ANOVA) was performed using the G*Power 3.1.9.7. program. The final number of participants recruited was 23, 22, and 22 in Experimental groups 1, 2, and 3, respectively. And all the participants completed a questionnaire in full and joined in the intervention, and all of their data were included in the analyses.

### 2.3. Research Variables

#### 2.3.1. Attitude toward Smoking Cessation Interventions

To measure attitudes toward smoking cessation interventions, we used the instrument developed by Macnally et al. [20], which was validated through translation and reverse translation by Choi and Kim [21]. The instrument consists of eight questions, with responses ranging from 1 point for “not at all” to 5 points for “very much so” and a score range of 8–40 points. Higher scores indicate a more positive attitude toward smoking cessation intervention. In the study by Macnally et al. [20], the Cronbach’s α value of the reliability of the tool was .84, and in the study by Choi and Kim [21], the Cronbach’s α value was .88, whereas, in the present study, it was .78.

#### 2.3.2. Self-Efficacy of Smoking Cessation Interventions

To measure the self-efficacy of smoking cessation interventions, an instrument developed by Sohn et al. [22] and modified and supplemented by Myeong-Hee Song et al. [23] was used. It consists of nine questions, with responses ranging from 1 point for “not at all” to 5 points for “very much so” and a score range of 9–45 points. Further, higher scores indicated higher self-efficacy of smoking cessation interventions. In the study by Choi and Kim (2018) [21], the Cronbach’s α value of the tool was .85, and it was .90 for this study.

#### 2.3.3. Intention to Deliver Smoking Cessation Intervention

To measure the intention to deliver smoking cessation intervention, an instrument developed by Segaar et al. [24] and modified and supplemented by Choi et al. [21] was used. It comprises three questions, with responses ranging from 1 point for “not at all” to 5 points for “very much so” and a score range of 3–15 points. Higher scores indicated higher intention to fulfill smoking cessation interventions. In Choi and Kim’s [21], the reliability Cronbach’s α value of the tool was .83, and it was .78 for the present study.

### 2.4. Research Process

#### 2.4.1. Researcher Preparation

The researcher (first author) majored in adult nursing and had experience developing smoking prevention and smoking-cessation-education-related research, lectures, and educational programs over the past 25 years. Additionally, she has experience in developing and distributing counseling programs for medical personnel and has conducted numerous studies on smoking and smoking cessation.

#### 2.4.2. Smoking Cessation Counseling Education Program

(1)Smoking cessation counseling education module for pre-learning

The first educational material was a smoking cessation counseling education module for pre-learning to be used by all three groups. The module for pre-learning was presented to all participants immediately after the pre-test, and they were guided to complete it before the intervention stage. The pre-learning module was developed to teach participants about the strategies involving the stage of change (the 5A’s: Ask, Advise, Assess, Assist, and Arrange) and tailored counseling contents (the 5R’s: Relevance, Risk, Rewards, Repetition, and Roadblocks), national smoking cessation support institutions and services, and pharmaceutical drug therapy, which are considered as a core contents in delivering smoking cessation counseling.

Prior to each experimental group, participants were asked to form a sub-group composed of 2~3 individuals, and each sub-group was asked to investigate and discuss in order to answer the questions presented based on the module and to submit the results in the form of a report. In the module, six questions were presented to be answered through the scenario of a male patient who was hospitalized in the internal medicine ward through the emergency room due to worsening respiratory symptoms of chronic bronchitis corresponding to the pre-contemplation stage.

(2)Smoking cessation counseling education program

① Lecture

For the experimental group 1, the researcher provided a lecture with answers to each six questions presented in the pre-learning module. The group was guided to review the counseling samples presented in the module together in small groups and to complete their individual reflection journals.

② Online Video

Experimental group 2 received answers to questions on the pre-learning module same as Experimental group 1, and then received free online smoking cessation counseling expert education courses at the online anti-smoking education center of the National Smoking Support Center of the Korea Health Promotion Institute. The participants were asked to view a two-hour online video regarding smoking and health, understanding new types of tobacco, smoking cessation counseling techniques, smoking cessation counseling based on the principle of the 5A’s and 5R’s, smoking cessation behavior therapy, and smoking cessation drug therap.

③ Case-based peer role-play

Experimental group 3 received lecture on the pre-learning module as previous groups. For students to effectively perform the role play, a specific counseling guide developed based on the 5A’s and 5R’s was provided in advance, which they were asked to review. Then, the participants were asked to form a team of two, and the seven smoking scenarios developed based on the five change of stages of DiClemente et al. [25] were played, in which the roles of the smoker and counselor were alternately performed, the content was verified by a professional specialized in smoking cession, and feedback was provided. (Table 1).

#### 2.4.3. Pre-Test

A pre-test of the study variables was administered to all subjects two weeks prior to the start of the smoking cessation counseling education program to minimize the influence of memory of the study variables on the intervention and post-test. All data collections were administered by trained research assistants who were blinded to the participants’ allocation to the treatment groups.

#### 2.4.4. Post-Test

Post-test was conducted by pre-trained research assistant immediately after the application of the intervention for each group to prevent a third variable interaction as well as to prevent a maturation effect. The time required for the post-test was approximately 10 min. Further, as a reward, an online gift-card was provided to the participants after all the experimentation was completed for the groups.

### 2.5. Ethical Consideration

This study was conducted after receiving approval (IRB No 2-1040781-A-N-012021113HR) from the Institutional Research Ethics Review Committee of S University for the research content and methods. The participants listened to a sufficient explanation of the purpose and procedure of the study, participated in a question-and-answer session, and voluntarily signed a written consent form. It was explained that participation could be withdrawn at any stage of the study and no disadvantages would be incurred. Regarding research conducted during the clinical practice period, it was noted that it had no effect on the students’ practice grades. It was explained that the data provided during the study would guarantee anonymity and confidentiality, and would not be used for any purpose other than research.

### 2.6. Data Analyses

The collected data were analyzed using SPSS Windows software version 25.0. General characteristics were analyzed by real numbers and percentages, and the homogeneity of general characteristics and study variables between study groups were analyzed using the χ^2^-test, Fisher’s exact test, *t*-test, and one-way ANOVA. As for the normal distribution of the study variables, the Shapiro–Wilk test showed that all variables followed a normal distribution, and the effect of the smoking cessation counseling education program was analyzed using a repeated-measures ANOVA.

## 3. Results

### 3.1. Homogeneity Verification of Participants’ General Characteristics and Study Variables

Among the participants, 9 (13.4%) were smokers and 16 (23.9%) lived with smokers. In the pre-homogeneity test for study variables between each group, attitude toward smoking cessation intervention (F = 1.61, *p =* .207), self-efficacy in smoking cessation intervention (F = 0.82, *p =* .447), and intention to quit smoking intervention (F = 0.98, *p* = .383), there was no statistically significant differences (Table 2).

There were no statistically significant differences among the participants’ general characteristics in terms of research variables (attitude toward smoking cessation intervention, self-efficacy for smoking cessation intervention, and intention to deliver smoking intervention) (Table 3).

### 3.2. Effects of Attitude toward Smoking Cessation Intervention, Self-Efficacy for Smoking Cessation Intervention, and Intention to Deliver Smoking Intervention after Smoking Cessation Counseling Education Program

Table 4 and Figure 2 show the effects of the research variables according to the smoking cessation counseling education program for nursing students’ smoking cessation intervention. The details results are as follows.

The attitude toward smoking cessation intervention of all participants increased from 26.86 points before education to 30.68 points after education, showing a difference in the paired *t*-test (t = −3.47, *p* < .001). Within each experimental group, the differences were statistically significant in experimental group 2 (t = −2.48, *p* = .021) and experimental group 3 (t = −2.69, *p* = .013). As a result of the repeated-measures ANOVA, there was a statistically significant difference between the time points (F = 12.34, *p* < .001) and between the groups (F = 4.66, *p* = .013), on the other hand interaction effect of the group was not statistically significant (F = 1.50, *p* = .232).

Self-efficacy for smoking cessation intervention increased from 33.07 points before education to 36.28 points after education for all participants, showing a significant difference in the paired *t*-test (t = −3.49, *p* < .001). The repeated-measures ANOVA on the change in self-efficacy for smoking cessation intervention according to the measurement period between each group showed a statistically significant difference at each time point (F = 11.86, *p* < .001). There was no statistically significant difference between the groups (F = 1.32, *p* = .276) and interaction effect of group and time point (F = 0.05, *p* = .951).

Intention to deliver smoking cessation intervention increased from 10.81 points before education to 12.04 points after education for all participants, showing a difference in paired sample verification (t = −3.81, *p* < .001). Within each experimental group, the differences were statistically significant in the Experimental group 1 (t = −5.54, *p* < .001), experimental group 2 (t = −2.83, *p* = .010), and experimental group 3 (t = −3.49, *p* < .002). The repeated-measures ANOVA on the change in intention to deliver smoking cessation intervention according to the measurement period between each group showed a statistically significant difference at each time point (F = 14.26, *p* < .001). There was no statistically significant difference between the groups (F = 1.12, *p* = .334) and interaction effect of group and time point (F = 0.46, *p* = .636).

## 4. Discussion

This study was conducted to develop a smoking cessation counseling education program that can be used in undergraduate nursing education and verify its effectiveness in improving attitudes toward smoking cessation intervention, self-efficacy, and intention to deliver smoking cessation intervention.

The current smoking rate among nursing students in this study was 13.4%, which was lower than the 20.4% smoking rate among university students [26] and higher than the 11.4% smoking rate among women in their 20s in Korea [27,28,29]. It has been reported that nursing students who smoke are less likely to be perceived as role models for patient smoking cessation than nonsmoking nursing students [30], so for effective patient smoking cessation education, nursing students’ own smoking cessation should be a prerequisite. Although the smoking rate of the participants in this study was lower than that of women in their 20s, it is necessary to pay attention to smoking prevention and anti-smoking education in nursing major courses.

There was a statistically significant increase in attitudes toward smoking cessation interventions after participating in this study’s smoking cessation counseling education program in both Experiment 2 and Experiment 3 compared to Experiment 1. The mean scores of 29.86 ± 3.43 for Experiment 2, who participated in the online training program, and 30.68 ± 3.33 for Experiment 3, who participated in the case-based peer role-play, were somewhat lower than the mean scores of 31.79 ± 5.20 out of 40 in previous studies [21]. It has been shown that the more positive the attitude toward smoking cessation interventions, the higher the intention to engage in smoking cessation interventions [21]. So, the formation of positive attitudes toward smoking cessation interventions through smoking cessation counseling education programs will lead to an increase in intention to engage in smoking cessation interventions. In this study, attitudes toward smoking cessation interventions increased more in experimental group 3, which received a case-based peer role-play, than in experimental group 2, which received an online counseling education program, suggesting that experiential training with direct counseling contributed to improving nursing students’ attitudes toward smoking cessation interventions. Previous studies have reported that despite positive attitudes towards tobacco cessation education among healthcare providers in primary healthcare settings, inadequate tobacco cessation counseling was reported due to a lack of competency in practical tobacco counseling [31]. Therefore, in order to enhance positive attitudes toward tobacco cessation interventions and competence for effective tobacco cessation counseling interventions, practical case-based experiential education should be actively used in tobacco cessation counseling education programs for nursing students and health care workers rather than simple lecture type education.

Self-efficacy for smoking cessation interventions increased to 36.28 ± 5.45 for all students, a statistically significant difference, and was higher than the mean of 34.05 ± 5.10 out of 45 in a previous study of nursing students [21]. Although there were no statistical differences in self-efficacy for smoking cessation interventions after participating in the smoking cessation counseling education program in any of the groups, the greatest increase in self-efficacy for smoking cessation interventions was among students in Experimental group 2, which received the online smoking cessation counseling education program. In a previous study of nursing student tobacco cessation counseling knowledge and efficacy using online training and simulation, 110 nursing students reported gaining tobacco cessation counseling skills and expressed high levels of confidence [32]. Video education has been shown to be effective in changing health-related behaviors [33,34] and the online training program used in this study is a proven program used by the National Center for Tobacco Control, and it is believed that it provided positive modeling for nursing students by demonstrating knowledge and practical counseling examples. By observing and internalizing the successful counseling of experts through the online education program, nursing students developed a positive sense of efficacy for smoking cessation counseling, confirming the need for a standardized online education program that can be presented as a model for nursing students’ smoking cessation counseling education. In the future, the development of an online video education program that can be presented as a model for developing positive efficacy should be considered for nursing students’ smoking cessation counseling education programs [35].

In this study, the intention to deliver smoking cessation intervention was 12.04 ± 2.06 out of 15 after the smoking cessation counseling education program, which was higher than the deliver smoking cessation intervention of 11.76 ± 2.17 in a previous study [21] and 11.05 ± 2.37 in a study of psychiatric nurses [23], which may reflect the effectiveness of the education on smoking cessation intervention counseling and the negative attitudes toward smoking already formed by nursing students with little smoking experience. In all experimental groups in this study, deliver smoking cessation intervention was statistically significant as a result of the comparison of pre- and post-test, and it was confirmed that the application of smoking cessation counseling lectures alone could increase the intention to deliver smoking cessation intervention. Regardless of how tobacco cessation counseling is taught, trained healthcare providers with high intentions to implement tobacco cessation interventions are more likely to implement effective interventions in practice, which has been proven to help smokers quit. [9,23,36]. Recently, simulation-based training programs have been shown to increase participation in self-efficacy [22,37]. Case-based role-playing and simulation-based education for nursing students has been shown to be effective in various subjects [22,38,39], so it is a method that can be actively utilized for practical education, such as smoking cessation counseling. The development and application of a case-based simulated tobacco cessation counseling training program that can induce a variety of experiences in a tobacco cessation counseling training program for nursing students would be a strategic approach to strengthen the intention of tobacco cessation interventions. In this study, smoking cessation intervention intentions were found to be effective across all educational methods, suggesting that nursing education settings should adapt to the context of the educational setting rather than not implementing smoking cessation counseling education at all.

Healthcare providers have been shown to increase smoking cessation rates by as much as 30% with brief cessation counseling lasting less than 3 min, and health care providers who spend the most time in direct patient contact are important professionals who can have a significant impact on reducing smoking rates through tobacco cessation education [9,40,41]. Therefore, the implementation and utilization of tobacco cessation education programs within the regular curriculum of nursing schools preparing future healthcare professionals should be actively considered.

This study has nursing implicational significance in that it developed a learning module that can be utilized in actual nursing education settings to emphasize the importance of smoking cessation, which has been emphasized in the reality of the increasing incidence of chronic respiratory diseases. It is also significant in that it verified the effectiveness of smoking cessation counseling education for nursing students by confirming that when counseling education was conducted using the module, the subjects’ attitudes toward smoking cessation interventions, self-efficacy for smoking cessation interventions, and intention to deliver smoking cessation intervention increased.

However, this study has the following limitations. First, as this study was conducted during the clinical practice of nursing students at a major nursing in S city, the participants could not be randomly assigned to each experimental group. This study was a convenience sample of nursing students at a single university in a specific region, so it is difficult to generalize the results. Second, this study was conducted as part of a practicum course, so it is difficult to conduct a long-term study. Since only one follow-up test was conducted to check the effectiveness of the smoking cessation counseling education program, there is a limitation that the continuous effect of the smoking cessation counseling education program could not be checked. Third, the study was conducted in the form of case-based smoking cessation counseling practice through peer role-playing without using a standardized patient, which is used for nursing simulation practice training and evaluation.

Based on the findings of this study, the following suggestions are made. First, to confirm the effectiveness of the smoking cessation counseling module developed in this study, it is necessary to obtain a larger sample and conduct a repeated measures study on the dependent variable to verify the long-term effects. Second, it is necessary to compare the effectiveness of the smoking cessation counseling module using peer role-play with the educational effect of the smoking cessation counseling module using standardized patients. Third, we propose the development of a standardized case-based simulation program to increase the intention to implement smoking cessation interventions among nursing students who are underrepresented in education and practice. We also suggest that policy research should be conducted to actively promote the application of smoking cessation counseling education programs in regular curricula.

## 5. Conclusions

This study found that traditional lecture methods, a convenient online video program, and interactive case-based peer role-play were all effective in increasing nursing students’ intentions to deliver smoking cessation interventions. For future smoking cessation counseling education for nursing students, lecture, online video, and case-based peer role-play approaches should be selected in consideration of educational condition and environment, but the core content employed in the pre-learning module should be incorporated.

## Figures and Tables

**Figure 1 healthcare-11-02734-f001:**
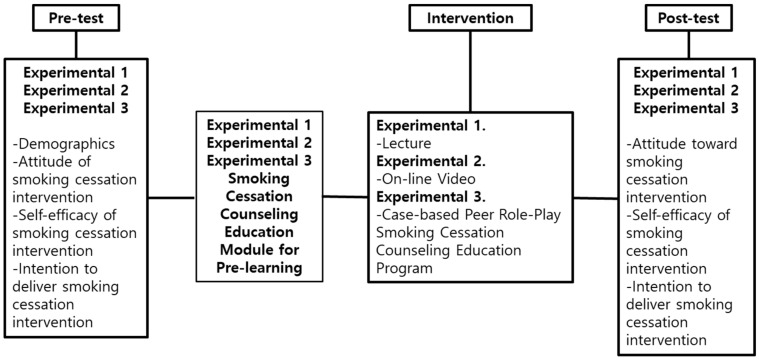
Research design: smoking cessation counseling education program.

**Figure 2 healthcare-11-02734-f002:**
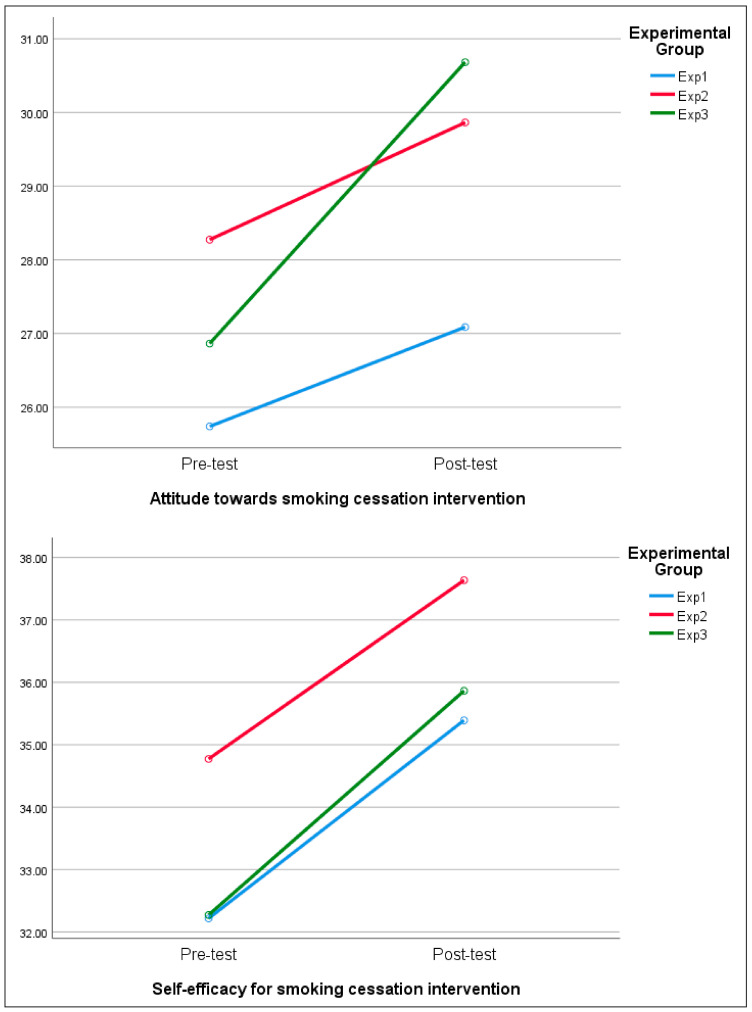
Effect of smoking cessation education program on attitude toward smoking cession intervention, self-efficacy for smoking cessation intervention, and intention to deliver smoking cessation intervention.

**Table 1 healthcare-11-02734-t001:** Developed case-base scenarios for peer role-play in this study.

No.	Stage of Change	Scenario
**1**	Precontemplation	A 55-year-old man visiting a health checkup center. High blood pressure, cholesterol, and triglyceride levels in hypertensive patients who are not interested in smoking cessation.
**2**	Precontemplation	A 36-year-old woman goes to a gynecology outpatient clinic for a prescription for oral contraceptives that she has been taking for the past 5 years. She is planning to become pregnant in the near future but is not yet ready and smokes half a pack of cigarettes a day, but does not believe her smoking habit will adversely affect her health.
**3**	Contemplation	A 56-year-old man presents to the outpatient department for shortness of breath and cough. The patient only knows that he should quit smoking because his wife is strongly encouraging him to do so.
**4**	Contemplation	A 38-year-old firefighter who visits a health center smoking cessation clinic and wants to discuss using a nicotine patch to reduce his tobacco use.
**5**	Contemplation	A 29-year-old woman who is flirting with her boyfriend and wants her non-smoker boyfriend to know about her smoking habits. There is the thought of quitting smoking and the lack of mental preparation.
**6**	Preparation	A 52-year-old man with a past history of essential hypertension. Has just moved from abroad due to a job promotion and needs to stop smoking immediately. His blood pressure is around 140/90 and is slightly overweight for his height (about 5 kg overweight).
**7**	Relapse	A 60-year-old man with a thrombosis in a right leg vein. Has been taking wafarin for 5 years and was admitted to hospital 1 year ago with shortness of breath and hypertension. Smoked about one pack of cigarettes per day for 30 years prior to admission, but has been smoke-free for about 1 year using nicotine patches on the recommendation of his physician and relapsed 2 weeks ago.

**Table 2 healthcare-11-02734-t002:** Homogeneity test of characteristics and dependent variables among groups.

Variables	Characteristics	Exp. 1 (n = 23)	Exp. 2 (n = 22)	Exp. 3 (n = 22)	χ^2^ or t(F)	*p*
		n (%) or M (S) ± SD	n (%) or M (S) ± SD	n (%) or M (S) ± SD		
Sex	Male	3 (25.0)	5 (41.7)	4 (33.3)	0.72	.675
	Female	20 (36.4)	17 (30.9)	18 (32.7)		
Age		23.67 ± 3.09			144.32	.837
Grade (GPA)	2.5–3.0	1 (33.3)	1 (33.3)	1 (33.3)	3.73	.753
	3.0–3.5	4 (25.0)	4 (25.0)	8 (50.0)		
	3.5–4.0	12 (38.7)	12 (38.7)	7 (22.6)		
	4.0–4.5	6 (35.3)	5 (29.4)	6 (35.3)		
Smoking status	Smoker	4 (44.4)	2 (22.2)	3 (33.3)	0.67	.902
	Non-smoker	19 (32.8)	20 (34.5)	19 (32.8)		
Residence	With Smoker	7 (43.8)	5 (31.3)	4 (25.0)	0.96	.678
	Without Smoker	16 (31.4)	17 (33.3)	18 (35.3)		
Attitude toward smoking cessation intervention		25.74 ± 4.30	28.27 ± 5.24	26.86 ± 4.64	1.61	.207
Self-efficacy for smoking cessation intervention		32.22 ± 7.51	34.77 ± 8.86	32.27 ± 6.24	0.82	.447
Intention to deliver smoking cessation intervention		10.39 ± 2.31	11.41 ± 2.89	10.64 ± 2.40	0.98	.383

Exp: Experimental group. Exp. 1 group: Smoking cessation education module + lecture. Exp. 2 group: Smoking cessation education module + online video education. Exp. 3 group: Smoking cessation education module + case-based peer role-play.

**Table 3 healthcare-11-02734-t003:** Characteristics and dependent variables according to the general characteristics.

Variables	Characteristics	n (%) or M ± SD	Attitude toward Smoking Cessation Intervention	Self-Efficacy for Smoking Cessation Intervention	Intention to Deliver Smoking Cessation Intervention
			t/F(p)	t/F(p)	t/F(p)
Sex	Male	12 (17.9)	0.71 (.479)	−0.08 (.937)	−0.71 (.481)
	Female	55 (82.1)			
Age		23.67 ± 3.09	144.32 (.837)	233.31 (.352)	97.71 (.271)
Grade	2.5–3.0	3 (4.5)	0.17 (.915)	0.72 (.543)	0.23 (.875)
	3.0–3.5	16 (23.9)			
	3.5–4.0	31 (46.3)			
	4.0–4.5	17 (25.4)			
Smoking status	Smoker	9 (13.4)	1.32 (.193)	0.82(.414)	0.36 (.720)
	Non-smoker	58 (86.6)			
Residence	With Smoker	16 (23.9)	−1.21 (.232)	0.22 (.828)	−1.00 (.320)
	Without Smoker	51 (67.1)			

**Table 4 healthcare-11-02734-t004:** Effect of research variables after smoking cessation counsel program.

Variables		Pre-Test	Post-Test	t	*p*	Source	F	*p*
		(M ± SD)	(M ± SD)					
Attitude toward smoking cessation intervention	Exp group 1. (n = 23)	25.74 ± 4.30	27.09 ± 3.59	−1.56	.134	T	12.34	<.001
	Exp group 2. (n = 22)	28.27 ± 5.24	29.86 ± 3.43	−2.48	.021	G	4.66	.013
	Exp group 3. (n = 22)	26.86 ± 4.64	30.68 ± 3.33	−2.69	.013	T × G	1.50	.232
	Paired-difference analysis	26.94 ± 4.78	29.18 ± 3.74	−3.47	.001			
Self-efficacy for smoking cessation intervention	Exp group 1. (n = 23)	32.22 ± 7.51	35.39 ± 4.57	−1.42	.170	T	11.86	<.001
	Exp group 2. (n = 22)	34.77 ± 8.86	37.64 ± 5.87	−2.06	.052	G	1.32	.276
	Exp group 3. (n = 22)	32.27 ± 6.24	35.86 ± 5.85	−1.16	.260	T × G	0.05	.951
	Paired-difference analysis	33.07 ± 7.59	36.28 ± 5.45	−3.49	.001			
Intention to deliver smoking cessation intervention	Exp group 1. (n = 23)	10.39 ± 2.31	11.61 ± 2.06	−5.54	<.001	T	14.26	<.001
	Exp group 2. (n = 22)	11.41 ± 2.89	12.27 ± 2.03	−2.83	.010	G	1.12	.334
	Exp group 3. (n = 22)	10.64 ± 2.40	12.27 ± 2.12	−3.50	.002	T × G	0.46	.636
	Paired-difference analysis	10.81 ± 2.54	12.04 ± 2.06	−3.81	<.001			

Exp: Experimental group. Exp. 1: Smoking cessation education module. Exp. 2: Smoking cessation education module + online video education. Exp. 3: Smoking cessation education module + case-based peer role-play. T: Pre–Post test. G: Smoking cessation education program. T × G: Time × group interaction effect.

## Data Availability

The datasets used and/or analyzed during the current study are available from the corresponding author upon reasonable request.

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
