# Peer review of "Effects of a Smoking Cessation Counseling Education Program on Nursing Students"

_healthcare, 2023, doi:10.3390/healthcare11202734_

Round 1
Reviewer 1 Report
Manuscript title: Effects of a Smoking Cessation Counseling Education Program in Nursing Students- Case-Based Peer Role-Play has some errors that must be addressed.
Abstract: Not written properly; results and conclusion are not clear. Which method is effective in teaching nursing students about smoking cessation programs? Keywords: simple and understandable
Introduction: In line 40 word “trnteined” does not make any sense. Explain leadership theories, like teaching methods most used in Korea and various smoking cessation programs. Authors have only explained the importance of Role play teaching techniques and explain others. Also, the need for study is not mentioned clearly.
Material and method: Too much elaboration; try to reduce the content.
Results: Well-written. The graphs and tables are self-explanatory and marked correctly.
Discussion: Too lengthy, concise it. Secondly, limitations are not mentioned in this study. What are the outcomes of the findings of this study? What should the curriculum-developing community add while planning the smoking cessation program?
Conclusion: Too elaborate, some points of conclusion should be added to the discussion.
References are correctly marked, and no duplication is reported.
Minor editing required
Author Response
Thank you for your critical review of this study. Please find attached the changes we have made in response to your review comments.

Reviewer 2 Report
Thank you for the opportunity to review the manuscript "Effects of a Smoking Cessation Counseling Education Program in Nursing Students_ Case-Based Peer Role-Play”. I appreciate you inviting me to provide feedback on this paper. The article addresses an important issue - training nursing students in smoking cessation counseling. This is valuable given nurses' role in helping patients quit smoking.
It was well-written and demonstrated a clear understanding of the subject matter. The results are promising, particularly for active role-play methods. Some limitations exist, but overall the study appears well-designed and implemented. However, I would like to raise a few concerns that emerged while reading the article. These concerns, I believe, are essential for further refinement:
1. The practical significance of the results is not expanded on. For example, how the level of improvements in attitudes, self-efficacy, and intentions might translate to actual clinical skills is not discussed.
2. Differences between the interventions are explained briefly but more detail unpacking why the role play in particular was most effective would add helpful interpretation.
3. Very few limitations of the study are acknowledged. The authors could better address biases, limitations in generalizability, lack of long-term follow-up
4. 5A's & 5R's need to be elaborate for the reader
Minor grammatical and typo editing require
Author Response

(The authors gave the same response as above.)

Reviewer 3 Report
The topic of the study is a prominent component in health sciences studies, and especially in nursing students who are future professionals with great responsibility and leadership in smoking cessation programs for smokers of all ages. So, the subject raised by the authors is an element of the nursing studies curriculum, forming an important part in the teaching of health education methods. For this reason, the study, in my opinion, is neither new nor of great scientific interest. Moreover, it is also well known that participatory teaching methodologies such as "Case-based Role-Play" have a more effect than traditional theoretical classes, with numerous evidences.
Furthermore, the sample is very small (in each intervention group there are less than 25 participants) and although there are positive effects especially in group 3, they were measured only once, and there are no medium-term results of the intervention.
Other comments:
The intervals of the "Grade" variable in tables 2 and 3 are repeated, or they are not well understood, this variable may be analyzed by semesters....
There is no explanation or comment on the sample size in the limitation section.
Author Response

(The authors gave the same response as above.)

Round 2
Reviewer 1 Report
Authors have addressed all the comments.
No editing is required
Author Response
Thank you for agreeing to the researcher's response to the Round 1 review.
Reviewer 3 Report
My appreciation to the authors for considering the changes.
Comments:
In the introduction, the information added "Lecture is the most commonly used and known as the cost-effective method of teaching in nursing education. Recently various on-line education program has been developed and are widely used in nursing education because they allowed for repeated learning based on the needs of the students". Also, the information in the last paragraph of the introduction is not referenced in detail: “In this study, .......” The authors should indicate which study they are referring to. What reference 12 or 13 or 14?
Finally, I agree with the rest of the changes made in the manuscript.
Author Response
Thank you for agreeing to the researcher's response to the Round 1 review. We will respond to your additional comments.
